# Increased Neural Efficiency in Visual Word Recognition: Evidence from Alterations in Event-Related Potentials and Multiscale Entropy

**DOI:** 10.3390/e23030304

**Published:** 2021-03-04

**Authors:** Kelsey Cnudde, Sophia van Hees, Sage Brown, Gwen van der Wijk, Penny M. Pexman, Andrea B. Protzner

**Affiliations:** 1Department of Psychology, University of Calgary, Calgary, AB T2N 1N4, Canada; sophia.vanhees@gmail.com (S.v.H.); sabrown@ualberta.ca (S.B.); gwen.vanderwijk@ucalgary.ca (G.v.d.W.); pexman@ucalgary.ca (P.M.P.); protzner@ucalgary.ca (A.B.P.); 2Hotchkiss Brain Institute, University of Calgary, Calgary, AB T2N 4N1, Canada; 3The Mathison Centre for Mental Health Research & Education, University of Calgary, Calgary, AB T2N 4Z6, Canada

**Keywords:** visual word recognition, lexical decision task (LDT), brain signal complexity, multiscale entropy, event-related potential (ERP)

## Abstract

Visual word recognition is a relatively effortless process, but recent research suggests the system involved is malleable, with evidence of increases in behavioural efficiency after prolonged lexical decision task (LDT) performance. However, the extent of neural changes has yet to be characterized in this context. The neural changes that occur could be related to a shift from initially effortful performance that is supported by control-related processing, to efficient task performance that is supported by domain-specific processing. To investigate this, we replicated the British Lexicon Project, and had participants complete 16 h of LDT over several days. We recorded electroencephalography (EEG) at three intervals to track neural change during LDT performance and assessed event-related potentials and brain signal complexity. We found that response times decreased during LDT performance, and there was evidence of neural change through N170, P200, N400, and late positive component (LPC) amplitudes across the EEG sessions, which suggested a shift from control-related to domain-specific processing. We also found widespread complexity decreases alongside localized increases, suggesting that processing became more efficient with specific increases in processing flexibility. Together, these findings suggest that neural processing becomes more efficient and optimized to support prolonged LDT performance.

## 1. Introduction

Visual word recognition is the process of decoding a visual representation of a word and accessing its meaning, a human skill that is crucial to our ability to read and gain information from the world. Once we become literate, visual word recognition can be carried out quickly and automatically [1]. These qualities might suggest that visual word recognition is a relatively effortless process. However, when faced with the standard word recognition paradigm, a yes–no lexical decision task (LDT; is the stimulus a word?), some degree of thought and attention is required before a decision can be made, in order to evaluate whether the letter string is familiar or unfamiliar. In this way, visual word recognition as part of LDT performance is not initially efficient. Interestingly, the processes involved in LDT performance may have the potential to become more efficient after prolonged task performance. For example, findings of the British Lexicon Project (BLP; [2]; see also the Dutch Lexicon Project [3]), suggest that the efficiency of visual word recognition can improve. In the BLP, the authors found that LDT response times (RT) decreased across a prolonged period of LDT performance during which participants made lexicality decisions about a large portion of the lexicon (around 14,000 words).

In research examining processes involved in the shift from effortful to more efficient task-related processing, a shift from control-related to domain-specific neural processing has been found. For example, Jansma et al. [4] found reduced activation of working memory regions with automatic compared to controlled processing, as well as three behavioural effects indicative of automatic processing, namely reduced RT, reduced variance of RT, and increased accuracy [5]. Chein and Schneider [6] highlighted neural changes associated with the shift from controlled to automatic processing in terms of the dual-processing framework associated with theories of learning. The dual processing framework proposes that early, effortful task performance is supported by central neural resources that assist controlled processing, such as working memory, attention, and performance monitoring. As performance improves, local associations are made that enable more efficient processing of task-specific information; support from central resources decreases, so that task performance becomes reliant mostly on task-specific processing [7,8,9,10,11,12,13]. Indeed, in both a meta-analysis and a separate experiment, Chein and Schneider [6] found this pattern of neural change from controlled to more automatic processing, such that initial neural activity was primarily associated with cognitive control processes and shifted to more domain-specific processing as task performance became more automatic.

An examination of the neural changes involved in the shift from effortful to more automatic and efficient performance in LDT has not yet been undertaken, but previous research examining different types of lexical expertise has identified relationships between behavioural differences and neural effects. For example, Scrabble experts, who have extensive experience with lexical tasks [14,15], showed increased activation in bilateral working memory and visual perceptual regions during LDT, while age-matched controls showed more typical activation in language regions [16]. In electroencephalographic (EEG) work, Scrabble experts showed an increased late positive component (LPC) compared to controls when responding to vertically oriented stimuli in the LDT, possibly indicating improved stimulus evaluation and categorization afforded by experience [17]. The patterns of neural differences found in these studies contrast those by Jansma et al. [4] and Chein and Schneider [6], as instead of less reliance on control-related processes with what is presumed to be more efficient task performance afforded by experience, these results suggest greater reliance on control-related processes with more efficient processing. However, contrasts in study design may account for some of these differences, as the evidence from Scrabble experts comes from between-subjects comparisons, and so limits the conclusions we can draw about the nature of experience-related changes in lexical processing.

As behavioural evidence from the BLP [2] indicates that performance on the LDT can become more efficient with prolonged task performance, further investigation is warranted to determine what pattern of neural changes occurs. That is, does the brain support prolonged task performance through a shift from control-related to more domain-specific processing, or does control-related processing remain prominent and possibly aid more efficient processing? To investigate this, we utilized a within-subjects design and tracked behavioural changes across 16 h (spread over six to 10 days) of LDT performance and tracked neural changes with EEG measurement at three timepoints. To characterize the changes that occur with prolonged LDT performance and more efficient processing, we focused on two measures of brain function: event-related potentials (ERPs), and brain signal complexity, which reflects processing capacity in the brain [18,19].

The LDT is a common visual word recognition task that is used to probe aspects of visual word recognition and examine the influence of different lexical characteristics on lexical processing. In addition, the LDT has application beyond the context of lexical research. Performance of the LDT requires input from cognitive processes that support visual word recognition, but also requires more general cognitive processing, including attention, stimulus evaluation, and cognitive resource allocation, which all represent control-related processes. By using the LDT, our research helps inform specific aspects of lexical research, but importantly, it also has implications for a broader context. In particular, it can serve as a model for how neural processing becomes more efficient with prolonged task performance.

Based on processes associated with skill acquisition and learning, as well as processes proposed by the dual-processing framework, neural changes that accompany the behavioural changes found with prolonged LDT performance and increased efficiency of processing could occur through a shift from control-related to domain-specific processing. In general, when beginning to perform a new skill or task, performance is effortful. Although literate adults already have expertise in identifying words, the LDT is not an everyday task, so performance initially will be less than optimal and supported by a neural network that responds to novel task demands [20]. Drawing from dual-processing theory, this early neural network will involve cognitive control processes [6]. As task performance becomes more efficient, the network supporting task performance will become more specific and optimized [20,21,22], and will rely less on control processes and more on domain-specific processing [6]. In turn, with increased processing efficiency, this task network will enable LDT performance to become more automatic.

We expected to find changes in the ERP components associated with visual word recognition and LDT performance, namely orthographic processing, attention, semantic processing, stimulus evaluation, and categorization [23]. These are reflected in the N170, P200, N400, and LPC. The N170, an early negative component in occipital-temporal electrodes, has been associated with visual orthographic processing, with greater amplitudes observed in the left hemisphere for orthographic compared to nonorthographic stimuli [23,24,25,26]. The P200 is an early positive component, peaking between 150 and 300 ms in frontal and central electrodes. The P200 is often considered in the context of attentional processing, where increased attention is associated with decreased P200 amplitudes in various task conditions [27,28].

The N400 has been related to the processing of meaning, with variable topography depending on the task and stimuli used (see [29] for a review). Several studies show N400 effects in frontal scalp sites using lexical stimuli with, for example, concreteness and lexical manipulations [30,31], ambiguity [32], repetition [29], and attention and semantic priming [33,34,35]. N400 effects exist in a range of contexts, so this component could reflect processes related to lexical access, where reduced amplitudes are associated with easier lexical access [33,36], or more generally, with ease of processing stimulus meaning, with reduced amplitudes representative of facilitated semantic processing [29]. 

Lastly, stimulus evaluation and categorization have been associated with a late positive component (LPC) that peaks around 600 ms post-stimulus in central-posterior electrodes [37,38]. LPC amplitude decreases for task conditions requiring fewer attentional or cognitive resources, such as for congruent word pairs in priming studies [39,40], categorically and evaluatively consistent stimuli [37], and processing of concrete versus abstract words [30,41,42].

With prolonged LDT performance and increased efficiency of visual word recognition, we expected to see changes in these components in relation to a shift from controlled to domain-specific processing. More specifically, we expected that ERPs related to control processes, P200 and LPC, would show increases early on due to the proposed initial reliance on control processes. As LDT performance becomes more efficient, we expected P200 and LPC effects to decrease, represented by increased P200 and decreased LPC amplitudes, as control processes become less essential. Conversely, because the N170 and N400 are associated with more domain-specific processing, these ERPs should be weaker at the start of the task but should increase as task efficiency improves and domain-specific processing becomes more prominent.

As a second measure of brain function associated with increased efficiency resulting from prolonged LDT performance, we examined brain signal complexity. The temporal fluctuations in brain signal have both stochastic and deterministic properties and thus are neither completely predictable nor entirely random. This structural richness should be thought of as *complex* rather than simply variable [43,44], with complexity enabling the formation of and transition between different network configurations [45,46]. As such, brain signal complexity reflects the information processing capacity of the brain and the repertoire of potential responses that can be produced [18,46]. Changes in complexity can provide important information on *how* neural changes related to prolonged LDT performance may occur. We evaluated changes in signal complexity using multiscale entropy (MSE), which measures the way signals behave over a range of temporal scales from fine (e.g., over 2 ms intervals) to coarse (e.g., over 40 ms intervals) [18,43,44]. 

In terms of lower (i.e., more regular) brain signal complexity, Burles et al. [47] found that lower complexity in frontal electrodes was associated with faster and more accurate responses during a mental rotation task. The authors suggested that the lower complexity represented decreased cognitive load in frontal brain regions, which facilitated responding [47]. Heisz et al. [48] also reported decreased signal complexity and faster RTs with stimulus repetition. This decrease in neural response with stimulus repetition, often referred to as repetition suppression, can be explained in several ways [48] (for a review, see [49]). For example, it could be due to decreased firing rate of neurons in the task network (i.e., the fatigue model) [50,51]; extinction of response from neurons less essential in identifying the stimulus, with overall fewer neurons firing in the task network (i.e., the sharpening model) [52,53]; or faster stimulus processing, through shorter latencies or durations of neural firing within the task network (i.e., the facilitation model) [54]. Therefore, decreased signal complexity may indicate increased efficiency in processing. Taking the sharpening model for example, with fewer, more essential neurons responding to the repeated stimulus, decreased neural response could facilitate more efficient processing [49], as the task network becomes more specific and optimized.

In the current study, we replicated the BLP [2], in which participants completed 16 h of LDT performance over an average period of one week. In order to track the neural changes associated with visual word recognition, we employed EEG to measure neural response at three time points throughout LDT performance: first, at the beginning of task performance to gain a baseline measure of neural response; second, just prior to midway through task performance, as this was when the original BLP showed RT decreases began to stabilize; and third, at the end of task performance to gain a final measure of neural response. We expected to replicate the behavioural findings of the BLP, with a response time decrease between the first and second EEG time points, and a leveling off of the response time decrease between the second and third EEG time points. We also expected to find evidence of a shift from control-related processing early on during LDT performance, to more domain-specific processing as LDT performance becomes more efficient. If this shift occurs, we expected it would be evident through increased P200 and decreased LPC amplitude, and increased N170 and N400 amplitude over time. Lastly, we expected to see brain signal complexity decrease across the time points, which would represent an increase in neural efficiency.

To foreshadow the key contributions of the present study, we found evidence of both behavioural and neural alterations with prolonged LDT performance. Across the three EEG sessions, participants’ RTs on LDT trials decreased, replicating the findings of the BLP, and suggesting an increase in behavioural efficiency occurred. Consistent with this finding, ERP changes indicated a role for control-related processing early in the task; later in the task, we observed increased ERPs associated with domain-specific processing. To our knowledge, this is the first study to investigate the nature of changes in brain signal complexity with prolonged task performance. The complexity changes that occurred largely suggest that neural processing became more efficient, as widespread decreases in MSE were found. However, unexpectedly, we also found evidence of specific increases in MSE, suggesting that the increase in efficiency was accompanied by specific increases in processing flexibility.

## 2. Materials and Methods

### 2.1. Participants

Twenty-two healthy adults (11 females, 11 males) participated as part of a larger study. Inclusion criteria were right-handedness, English as a first language, and normal or corrected-to-normal vision. Exclusion criteria included a history of neurological disease or disorder, mental illness, head trauma, alcohol or drug abuse, or use of psychotropic medications within the last two years. Data from two participants were excluded from our analyses, one for failing to maintain sufficient accuracy, and another for failing to complete all LDT blocks. The remaining participants (10 females, 10 males) ranged in age from 20 to 28 years (*M* = 23.70, *SD* = 2.77), and had 13 to 23 years of education (*M* = 17.15, *SD* = 2.37). Ethics approval was obtained from the Conjoint Faculties Research Ethics Board of the University of Calgary (REB14-2419), and the study was conducted in accordance with the Declaration of Helsinki. All participants provided written informed consent before taking part in the study and were provided with monetary compensation for their participation.

### 2.2. Stimuli

Word stimuli were obtained from the BLP [2]. From the total list of 14,365 word stimuli, we selected words that were 3–8 letters in length and elicited BLP response accuracy greater than 10%. These 10.000 word stimuli were used to generate the same number of nonwords using Wuggy, a pseudoword generator [55]. Word and nonword stimuli were divided into 40 blocks of 500 items (250 words, 250 nonwords). Blocks were matched for word length, morphological structure, syllabic and subsyllabic structure, and transition frequencies of the subsyllabic segments. Monosyllabic nonwords differed from target words on one subsyllabic segment (onset, nucleus, or coda), whereas disyllabic nonwords differed on two subsyllabic segments. Words were matched across blocks for frequency (British National Corpus), orthographic neighbourhood density [56], orthographic Levenshtein distance [57], number of letters, and number of syllables.

### 2.3. LDT Procedure

LDT performance was spread over 16 h within a one-week period, and participants were asked to maintain minimum 80% accuracy. Each trial began with the presentation of a central fixation cross on a computer screen for 250–750 ms (jittered around *M* = 500 ms), followed by the stimulus, presented in uppercase white typeface on a plain black background. Participants responded as quickly and accurately as possible, pressing ‘D’ with their left index finger for a nonword response, or ‘K’ with their right index finger for a word response. The stimulus remained on screen until a response was made, followed by a 1000 ms blank screen, and then the start of the next trial. Self-paced break screens appeared every 100 trials. At the end of each block of 500 trials, participants received accuracy feedback. Participants completed nine blocks with simultaneous EEG recording (described below) and completed the remainder of the blocks online, outside of the lab via Qualtrics (12 blocks between EEG Sessions 1 and 2, 19 blocks between EEG Sessions 2 and 3). Order of blocks was randomized for each participant. Only accuracy was recorded for online blocks, as differences in internet connection speeds may have influenced RTs.

### 2.4. EEG Procedure and Preprocessing

Participants completed LDT with simultaneous EEG recording at the beginning (Blocks 1 to 3), just prior to midway through (Blocks 16 to 18), and at the end of task performance (Blocks 38 to 40). Stimuli were presented on a 24-inch Hewlitt-Packard (HP) lp2475w monitor using Presentation Software (Version 16.1, Neurobehavioural Systems, Inc., Berkeley, CA, USA). Participants were seated approximately 80 cm from the monitor in a dimly lit, radio frequency shielded, and sound attenuated chamber. Visual angle ranged from 2.4 × 0.6° for three-letter strings to 6.4 × 0.6° for eight-letter strings. RT and accuracy were recorded in addition to EEG.

EEG was recorded from 64 electrodes (Cz as reference) at a 500 Hz sampling rate with a band-pass of 0.05–100 Hz. An EasyCap (10/20 positioning) and Brain Vision actiCHamp system with active electrodes (Brain Products GmbH, Gilching, Germany) was used, and all electrode impedances were below 17 kΩ at the start of recording. Data were preprocessed in EEGLAB (Version 14.1.2) [58]. Raw data were band-pass filtered at 0.1–55 Hz, noisy channels were interpolated, and all data were re-referenced to an average reference. Artefact removal was completed using independent component analysis, and components carrying muscle or ocular artefacts (i.e., eye blinks, saccades, horizontal eye movements) were removed. Data were segmented into epochs from 200 ms pre-stimulus onset to 1000 ms post-stimulus onset and baseline corrected to the 200 ms pre-stimulus interval. Finally, trials were visually inspected and any remaining trials with excessive noise were removed.

### 2.5. Behavioural Analyses

We analyzed word and nonword trials separately to directly compare with the BLP results [2], using 2 (lexicality: word, nonword) by 3 (EEG session: Session 1, Session 2, Session 3) repeated measures ANOVAs for RT and accuracy. For any resulting significant interactions, follow-up analyses of simple main effects were conducted with Bonferroni correction (α = 0.017).

### 2.6. Electrophysiological Analyses

We analyzed brain data for word and nonword trials together to gauge overall changes related to prolonged LDT performance. We expected that the cognitive processes affected would be similar for words and nonwords, as the diffusion model [59,60] suggests that both are evaluated according to the same processes during lexical decisions. Correct trials for all stimuli were combined and analyzed in two ways. First, to facilitate analysis of ERP and MSE changes across the three EEG sessions (i.e., Session 1 vs. 2 vs. 3), trials for each participant for each of the three EEG sessions were averaged within participant, providing measures of the average neural response for each EEG session.

Second, we analyzed ERP and MSE changes within an EEG session by dividing a session into 10 consecutive segments, each containing 10% of trials. For example, if a participant had 950 correct trials in EEG Session 1, Segment 1 would contain trials 1–95, Segment 2 would contain trials 96–190, etc. If the total number of correct trials was not evenly divisible by 10, the remainder was added by 1 to each segment sequentially until no trials remained. This provided measures of the average neural response for each segment within an EEG session and enabled the examination of changes within each EEG session. The within session analyses were Bonferroni corrected at α = 0.008.

#### 2.6.1. Event-Related Potentials (ERPs)

ERPs were computed for each participant at each electrode using the ERPLAB toolbox (Version 7.0.0) [61] within the EEGLAB software (Version 14.1.2) [58] available for MATLAB (R2014a).

#### 2.6.2. Multiscale Entropy (MSE)

Multiscale entropy (MSE) was used to estimate brain signal complexity and was calculated in MATLAB (R2014a) using the algorithm available at https://www.physionet.org/content/mse/ (accessed on 26 October 2018) [18,43,44]. A detailed description of MSE and its applicability in analyzing signal complexity is available in Costa et al. [43,44]. To summarize, the MSE algorithm calculates sample entropy as a measure of the predictability (or regularity) of the signal at different timescales. This happens in two steps. First, data points within non-overlapping windows were averaged together to create 22 timescales. For example, the original time series was Scale 1 (i.e., 2 ms windows in the context of our 500 Hz sampling rate), Scale 2 averaged over two time points (i.e., 4 ms windows), up until Scale 22 (i.e., 44 ms windows). Second, sample entropy was calculated for each timescale by evaluating the probability of finding repetitive patterns based on pattern length *m* = 2 and similarity criterion *r* = 0.50 [18]. The pattern length means two consecutive data points were used for pattern matching; sample entropy therefore reflects the probability that two sequences that matched on the first two data points also matched on the next data point. The similarity criterion means that for two data points to be considered matching, the absolute amplitude difference between the two data points should be less than or equal to 50% of the original time series standard deviation. Electrode-specific MSE was calculated for each participant on single trials and then averaged across all trials within a given condition (i.e., an EEG session, or a segment of an EEG session).

#### 2.6.3. Partial Least Squares (PLS) Analysis

To analyze the ERP and MSE data, task partial least squares (PLS) analysis was used in MATLAB (R2014a; http://www.rotman-baycrest.on.ca/index.php?section=345; accessed on 21 November 2018) [18,62,63,64]. PLS is a data-driven multivariate analysis technique that operates on the entire data structure at once, identifying patterns of maximal covariance between ERPs/MSE and conditions across all electrodes and timescales simultaneously. Task PLS is similar to principal components analysis in that *a priori* contrasts are not specified. Instead, latent variables (LVs) showing similarities or differences between experimental conditions are identified. LVs are extracted in order of highest to lowest amount of covariance explained between conditions and neural activity. Each LV contains three vectors: design saliences identify a contrast of the similarities and differences between conditions; electrode saliences identify a particular pattern of electrodes and timescales that are most related to the condition differences; the singular value represents the strength of the effect expressed by the LV, i.e., the proportion of covariance accounted for.

Statistical testing was performed at two levels. First, the overall significance of the LV was calculated using permutation tests to assess whether the effect represented in a given LV is strong enough to be considered different from random noise. The order of conditions for each subject were reassigned (with the order of subjects remaining fixed), and PLS was recomputed on the permuted data. We ran 500 permutations so that a probability value would be derived from the number of times out of 500 that the singular value from each permuted data set is greater than or equal to that of the original data; if this probability is very low (*p* < 0.05), the LV is considered significant. Second, the stability of the identified pattern over participants was determined through bootstrap resampling, in which subjects were reassigned to conditions (with the order of conditions remaining fixed). The pattern is considered to be reliable if the electrode salience value is not dependent on which combination of subjects are included in each sample and is quantified by the bootstrap ratio (element loadings/standard error of the bootstrap distribution for each element). Bootstrap ratios are similar to *z*-scores but should be interpreted as confidence intervals. We used 500 bootstrap samples and a minimum threshold of 2.0, which corresponds to a 95% confidence interval or *p* value < 0.05. In some analyses, we used a higher bootstrap threshold as necessary to best illustrate the effects, which corresponds to an even lower *p* value. As each statistical test is computed in one mathematical step, no correction for multiple comparisons is necessary.

### 2.7. Code Availability

The code used for the current analyses is available on the Open Science Framework website (https://osf.io/da478/, accessed on 28 December 2020).

## 3. Results

All participants completed the LDT blocks in six to 10 days (*M* = 6.75, *SD* = 1.07). For all analyses, trials with RT ± 2.5 SD from the mean of each condition for each participant were excluded (2.5% of all data). For RT and neural analyses, trials with incorrect responses (12.3% of all trials) were additionally excluded.

### 3.1. Behavioural Results

RT and accuracy analyses (two-tailed, α = 0.05, unless otherwise indicated) were conducted in R (Version 1.1.456) using “ez” [65] and “rstatix” [66] packages. Normality and sphericity assumptions were met (Shapiro–Wilk and Mauchly’s *p* > 0.05, respectively) unless otherwise stated. Sphericity violations were corrected with Greenhouse–Geisser. Table 1 provides RT and accuracy means and standard deviations for the three EEG sessions.

#### 3.1.1. Response Time

There was a significant main effect of lexicality, *F*(1, 19) = 25.33, *p* < 0.001, η^2^_G_ = 0.08, a significant main effect of EEG session, *F*(2, 38) = 10.90, *p* < 0.001, η^2^_G_ = 0.12, and a significant interaction between lexicality and EEG session, *F*(1.19, 22.65) = 6.27, *p* =0.016, η^2^_G_ = 0.01 (Greenhouse–Geisser correction). To parse this interaction, word and, separately, nonword RTs for the three EEG sessions were analyzed (see Figure 1a). For words, there was a significant main effect of EEG session, *F*(2, 38) = 7.00, *p* = 0.003, η^2^_G_ = 0.08. Follow-up paired samples *t*-tests (α = 0.017) indicated a significant decrease in RT from Session 1 to Session 3, *t*(19) = 3.58, *SE* = 23.43, *p* = 0.002, *d* = 0.80, with nonsignificant differences between Sessions 1 and 2, *t*(19) = 1.80, *SE* = 26.22, *p* = 0.088, *d* = 0.40, and between Sessions 2 and 3, *t*(19) = 2.19, *SE* = 16.77, *p* = 0.041, *d* = 0.49. For nonwords, there was a significant main effect of EEG session, *F*(1.46, 27.79) = 12.14, *p* < 0.001, η^2^_G_ = 0.15 (Greenhouse–Geisser correction). Follow-up paired samples *t*-tests (α = 0.017) indicated a significant decrease in RT from Session 1 to Session 2, *t*(19) = 2.84 *SE* = 35.30, *p* = 0.011, *d* = 0.63, and from Session 1 to Session 3, *t*(19) = 4.31, *SE* = 33.91, *p* < 0.001, *d* = 0.96, but not between Sessions 2 and 3, *t*(19) = 2.41, *SE* = 19.11, *p* = 0.026, *d* = 0.54.

#### 3.1.2. Accuracy

Normality was violated in three data subsets: Session 1 nonwords, *W*(19) = 0.82, *p* = 0.002; Session 2 nonwords, *W*(19) = 0.85, *p* = 0.005; and Session 2 words, *W*(19) = 0.88, *p* = 0.015. However, as there is no appropriate nonparametric alternative, and in order to compare these results with the BLP, we maintained use of the parametric factorial repeated measures ANOVA. There was a significant main effect of lexicality, *F*(1, 19) = 34.06, *p* < 0.001, η^2^_G_ = 0.41, where responses to nonwords were more accurate than to words (*M* = 0.93, *SE* = 0.01 vs. *M* = 0.86, *SE* = 0.01; see Figure 1b). The main effect of EEG session (*F*(2, 38) = 0.74, *p* = 0.484, η^2^_G_ = 0.003) and the interaction between lexicality and EEG session (*F*(1.39, 26.35) = 2.26, *p* = 0.138, η^2^_G_ = 0.02; Greenhouse–Geisser correction) were both nonsignificant.

### 3.2. Event-Related Potentials

#### 3.2.1. Across Session ERPs

The task PLS analysis resulted in one significant LV (*p* < 0.001; Figure 2), which differentiated between Session 1 versus Session 2 and 3, with no difference between Sessions 2 and 3. The analysis highlighted four ERP components: the N170, P200, N400, and LPC. The N170 amplitude became more negative from Session 1 to Sessions 2 and 3. This difference was stable between 170–210 ms in bilateral occipito-parietal electrodes. The P200 and N400 amplitudes became more positive from Session 1 to Sessions 2 and 3. For the P200, this difference was stable between 170–210 ms in bilateral frontal and fronto-central electrodes. For the N400, this difference was stable between 365–405 ms in bilateral frontal and fronto-central electrodes. The LPC amplitude became more negative from Session 1 to Sessions 2 and 3. This difference was stable between 550–1000 ms in bilateral centro-parietal and parietal electrodes.

#### 3.2.2. Within Session ERPs: Session 1

We next examined the changes that occurred over time during Session 1 (divided into 10 segments of approximately 100 trials each, for each participant) in more detail (α = 0.008). The task PLS revealed two significant LVs. The first LV (*p* < 0.001; Figure 3) differentiated between the beginning (Segments 1 and 2) and the end of Session 1 (Segments 8, 9, and 10). Four ERP components were highlighted by the analyses: the N170, P200, N400, and LPC. The N170 amplitude became more negative across the session. This difference was stable between 170–215 ms in bilateral occipito-parietal electrodes. The P200, N400, and LPC amplitudes became more positive across the session. These differences were stable between 170–215 ms in bilateral frontal, fronto-central, and central electrodes for the P200; 390–410 ms in left frontal and fronto-central electrodes for the N400; and 600–800 ms in bilateral centro-parietal and parietal electrodes for the LPC.

The second LV (*p* = 0.004; Figure 4) differentiated between the beginning and end of Session 1 (Segments 2, 3, and 9) and the middle of the session (Segments 6 and 7), and revealed that LPC amplitude decreased in the middle of the session, compared to the beginning and end. This difference was stable between 615–700 ms at mainly left central and centro-parietal electrodes. However, the results showed that the LPC amplitude increased overall within Session 1, due to the increase observed in the first LV (Figure 5a), and the increase from the middle to end of the session in the second LV (Figure 5b).

#### 3.2.3. Within Session ERPs: Session 2

The task PLS performed on the 10 segments of trials in Session 2 identified two significant LVs (α = 0.008). The first LV (*p* < 0.001; Figure 6) differentiated between the beginning (Segments 1 and 2) and middle of the session (Segments 5, 6, and 7). Two ERP components were highlighted: the N170 and P200. The N170 amplitude became more negative from the beginning to the middle of the session. This difference was stable between 175–275 ms in left occipito-parietal electrodes, and between 200–260 ms in right occipito-parietal electrodes. The P200 amplitude became more positive from the beginning to the middle of the session. This difference was stable between 185–220 ms in bilateral anterior-frontal, frontal, and fronto-central electrodes.

The second LV (*p* < 0.001; Figure 7) differentiated between Segments 1 and 6 compared to Segments 3, 8, 9, and 10. This analysis revealed a pattern of change where N170 amplitude during Segments 3, 8, 9, and 10 was more negative than during Segments 1 and 6. This difference was stable between 185–215 ms and was localized to bilateral occipital electrodes.

#### 3.2.4. Within Session ERPs: Session 3

The task PLS performed on the 10 segments of trials in Session 3 identified two significant LVs (α = 0.008). The first LV (*p* < 0.001; Figure 8) differentiated between the early to middle portion of the session (Segments 3, 4, 5, and 6) and the end of the session (Segments 7, 8, and 10). A negative slow wave was identified at central and left parietal and occipital electrodes. This effect became more negative later in the session, and was stable between 675–730 ms, followed by variable intermittent time ranges of stability. This component has similar topography and latency to the late posterior negativity (LPN; see discussion).

The second LV (*p* < 0.001; Figure 9) differentiated between the beginning (Segments 1 and 2) and middle/end of Session 3 (Segments 6, 7, 8, 9, and 10). Two ERP components were highlighted: the N170 and P200. The N170 amplitude decreased across the session. This difference was stable between 190–250 ms in bilateral occipito-parietal electrodes. The P200 amplitude increased across the session. This difference was stable from 190–240 ms in bilateral frontal and fronto-central electrodes.

### 3.3. Brain Signal Complexity

#### 3.3.1. Across Session MSE

The task PLS analysis resulted in one significant LV (*p* < 0.01; Figure 10). Similar to the ERP analysis, this LV differentiated between Session 1 versus Sessions 2 and 3. In later sessions (i.e., Sessions 2 and 3), fine to medium scale MSE (2–30 ms windows) decreased in right fronto-central and left posterior electrodes, as well as fine to coarse scale MSE decreases (10–44 ms windows) in a few bilateral frontal electrodes. In addition, there were localized increases in coarse scale MSE (30–44 ms windows) in Sessions 2 and 3, in right parietal electrodes.

#### 3.3.2. Within Session MSE: Session 1

The task PLS analysis performed on the 10 segments of trials in Session 1 (α = 0.008) resulted in one significant LV (*p* < 0.001; Figure 11), which differentiated between the beginning (Segments 1 and 2) and the middle/end of the session (Segments 6, 7, and 8). This pattern revealed that MSE decreased across the scalp towards the middle/end of the session in fine to medium scales (2–24 ms windows), and there were also a few localized decreases in coarse scale MSE (10–44 ms windows) in mainly central and right frontal, parietal, and occipital electrodes. In addition, there were a few localized increases in coarse scale MSE (36–40 ms windows) towards the middle/end of the session in central, centro-parietal, and temporo-parietal electrodes.

#### 3.3.3. Within Session MSE: Session 2

The task PLS analysis performed on the 10 segments of trials in Session 2 identified two significant LVs (α = 0.008; Figure 12). The first LV (*p* < 0.001) differentiated between the beginning (Segments 1 and 2) and middle of the session (Segments 5 and 6). This revealed widespread MSE decreases across the scalp from the beginning to the middle of the session, similar to the Session 1 results. Mid-scale decreases (10–30 ms windows) occurred mainly in left anterior and bilateral posterior electrodes, and coarse scale decreases (30–44 ms windows) were clustered in occipital and occipito-parietal electrodes. No MSE increases occurred.

The second LV (*p* = 0.002) differentiated between the beginning (Segments 1, 3, and 4) and end of the session (Segments 8 and 9). Across the session, MSE decreased in fine scales (2–20 ms windows) but increased in coarse scales (24–44 ms windows). The decreases were located mainly in left posterior electrodes, with additional effects in bilateral anterior electrodes. The increases occurred in bilateral frontal, central, and parietal electrodes.

#### 3.3.4. Within Session MSE: Session 3

The task PLS analysis performed on the 10 segments of trials in Session 3 identified one significant LV (α = 0.008; Figure 13). This LV (*p* < 0.001) differentiated between the beginning (Segments 1 and 2) and middle of the session (Segments 5 and 6). As in Sessions 1 and 2, a pattern of widespread MSE decreases emerged across the scalp from the beginning to the middle of the session. Mid-scale decreases (10–30 ms windows) occurred across the scalp, and coarse scale decreases (30–44 ms windows) occurred in a few bilateral anterior electrodes.

## 4. Discussion

In the present study, we explored changes in behavioural and neural efficiency associated with prolonged performance on a visual word recognition task. Through a replication of the BLP [2], participants completed 16 h of LDT, and we measured EEG on three occasions and obtained two measures of neural response: ERPs and signal complexity. As expected, we found evidence of previously reported [2,3] decreases in RT across sessions. We also found evidence of neural change through alterations in ERPs and MSE.

Behaviourally, RT decreased across the three EEG sessions, suggesting that prolonged LDT performance led to increased efficiency of visual word recognition. While our findings are generally consistent with those of the BLP [2], we did observe a slightly different pattern of RT change with task performance. For the nonword trials, we found a significant decrease from the first to second session, as well as from the first to third session, with RT plateauing between the second and third sessions. This is in line with the RT change found in the BLP. For the word trials, we found a significant decrease in RT from the first to third session, but not between the first and second session, perhaps suggesting that, for word trials, changes to visual word recognition happened slightly later in our study than in the BLP. In the analysis of response accuracy, we found very similar results to those of the BLP, as accuracy was higher for nonwords than for words throughout the experiment and remained relatively constant. However, accuracy to words in our study was slightly higher than for words in the BLP. This may partly be because we only presented words with BLP response accuracy greater than 10%, thus removing more difficult stimuli that would have reduced overall accuracy.

In their study on controlled versus automatic processing, Jansma et al. [4] highlighted three behavioural effects associated with more automatic processing: reduced RT, reduced RT variance, and increased accuracy [5]. With prolonged LDT performance, we observed two of these effects, namely, reductions in RT and reduced RT variance, which may suggest that LDT performance increased in efficiency and became more automatic as the sessions progressed. However, we did not find evidence of the third effect of increased accuracy, and instead found that accuracy for words and nonwords remained relatively stable across the sessions. Again, this could be due to the relative difficulty of the stimuli in our version of the LDT, and to the fact that participants were encouraged to maintain a particular level of accuracy (80%).

Overall, the greatest neural changes occurred early in LDT performance during the first EEG session, but more detailed views within each session suggest that changes continued to occur into the second and third sessions. In line with our expectations, these changes occurred in four ERP components: the N170, P200, N400, and LPC. Early effects within the first session revealed two patterns of change in the ERPs: a change from the beginning to the end of the session, which highlighted N170, P200, N400, and LPC effects, and also a change in LPC activity in the middle of the session compared to the beginning and end. Within the second session, analyses revealed N170 and P200 effects, mainly between the beginning and middle of the session. In the third session, N170 and P200 effects were also highlighted from the beginning to end of the session, in addition to an LPN effect from the middle to end of the session. The effects for each component will be described in more detail next.

The N170 amplitude increased with LDT performance, becoming more negative over time in bilateral occipito-parietal electrodes. This increase occurred from the first to second session and remained stable between the second and third sessions, but within session analyses also revealed that the N170 increased within each of the sessions. Previous research has associated the N170 with visual orthographic processing, based on findings that greater N170 amplitudes are produced for orthographic compared to nonorthographic stimuli [23,24,25,26]. However, one study found increased N170 amplitude in the LDT compared to a task focused on letter identification within a word, which suggests the N170 may also show sensitivity for lexical word properties, not just visual orthographic processing [67]. Behaviourally, Hargreaves et al. [68] found a smaller semantic (concreteness) effect in LDT for Scrabble experts, who have many years of experience with lexical tasks [14,15], compared to nonexperts, and inferred that Scrabble experts may emphasize orthographic processing to a greater extent during LDT than nonexperts. Similarly, our finding that N170 amplitude increased with prolonged LDT performance may reflect a change to greater reliance on visual orthographic processing when making lexical decisions, or perhaps greater reliance on lexical properties of words in general.

We found that the P200 amplitude increased across and within our three ERP sessions in bilateral fronto-central electrodes, becoming more positive with prolonged task performance. The P200 is often considered a marker of attentional processing, with decreased amplitudes being associated with increases in attention [27,28]. In relation to language, the P200 is related to sublexical processing, such as syllable parsing. Studies have found an inverse relationship between frontal P200 amplitude and the degree of supposed lexical activation that stems from early sublexical/syllable processing [28,69,70]. For example, words with higher-frequency initial syllables are associated with weaker P200 amplitudes, compared to words with low-frequency initial syllables, and are assumed to activate more of the lexicon during word recognition [69,71]. Conversely, greater P200 amplitudes were found when the colour boundary of a multicoloured word was mismatched from the syllable boundary, suggesting hindered syllable parsing and lexical activation [70]. Based on previous attentional research [27,28], the increased P200 amplitudes observed in the current study are consistent with the interpretation that the requirement for attentional processing decreases over time as LDT performance becomes more efficient. Alternatively, drawing from evidence of sublexical processing P200 effects [69,70,71], our results may indicate altered reliance on sublexical processing during visual word recognition. These possibilities should be considered in future research.

We also found that the N400 amplitude decreased with prolonged LDT performance, both within the first session and from the first to second session in bilateral frontal and fronto-central electrodes. The N400 has traditionally been related to semantic incongruity effects in centro-parietal electrodes (e.g., [72]). Frontal N400 effects also have been found with single word presentations, relating to a variety of conditions [29,30,31,32,33,34,35,36], so there is still uncertainty surrounding interpretation of the N400. However, processing of meaning seems central to N400 effects across contexts, so one hypothesis is that N400 effects reflect the ease of processing stimulus meaning, with decreased amplitude reflecting facilitated processing [29]. This is consistent with previous findings of altered semantic processing with LDT practice and lexical expertise. For example, in a reanalysis of the BLP dataset, effects of certain semantic richness variables (e.g., imageability) decreased across LDT blocks [73], and, as mentioned, reduced semantic effects on lexical decision times were found in Scrabble experts compared to controls [68]. One study also found reduced N400 amplitudes for processing onomatopoeic words compared to control words, suggesting increased ease of lexical access [36]. Based on these previous findings, the N400 effects in our study may suggest that increased LDT efficiency is associated with reduced semantic processing or facilitated lexical access.

The LPC showed a less consistent pattern of change in our analyses. Across the three sessions in bilateral centro-parietal and parietal electrodes, LPC amplitude decreased from the first to second session, and stayed consistent thereafter. Within the first session, however, two different patterns of change emerged. In the electrodes highlighted in the across session analysis, LPC amplitude increased over time within Session 1 (LV 1; see Figure 5a). In left central and centro-parietal electrodes, LPC amplitude initially decreased, but increased toward the end of the session (LV 2; see Figure 5b). Overall, the results suggest that LPC amplitude increased bilaterally during the first session, but this increase was more pronounced in bilateral centro-parietal and parietal electrodes than in left central electrodes. Previous research has linked the LPC to cognitive resource allocation and stimulus evaluation [37,38], with reduced amplitudes observed in conditions requiring fewer attentional or cognitive resources, whereas increased amplitudes are observed in conditions requiring greater attentional or cognitive resources [37,39,42]. Since the LPC amplitude increased within the first session, findings from previous research suggest there was greater reliance on attentional or cognitive resources during initial task performance, when the LDT was novel. However, as LPC amplitude decreased in later sessions, our results are consistent with the interpretation that fewer attentional and/or cognitive resources were required as performance became more efficient.

Within the third session, our analyses also highlighted a negative slow wave that became even more negative from the middle to end of the session in central and left parietal and occipital electrodes. Drawing from prior literature, this component has similar topography and latency as the late posterior negativity (LPN) that is often reported in memory studies [74,75], and may be indicative of action monitoring or stimulus evaluation processes [74,76]. More testing needs to be conducted to better understand the effect in the context of LDT.

Taken together, our ERP findings suggest that prolonged LDT performance was associated with decreased reliance on attentional processing and cognitive resource allocation, and increased reliance on orthographic processing, which is more domain specific. Our results are also consistent with a shift from more semantic processing to more orthographic processing with lexical experience [68]. In line with studies examining the shift from controlled to automatic processing [4,6], our ERP effects are consistent with a shift from reliance on cognitive control processes early in task performance to more domain-specific processing with increased task efficiency.

Our brain signal complexity analyses also revealed changes related to prolonged LDT performance; MSE mainly decreased across the scalp from the first session to the latter two sessions, except for a few localized increases. Across the three sessions, fine and coarse scale decreases occurred in bilateral frontal and central electrodes, and left posterior electrodes, while coarse scale increases occurred in right posterior electrodes. Within the first session, widespread MSE decreases occurred across the scalp, with specific coarse scale increases in mainly right centro- and temporal-parietal electrodes. Within the second session, fine, mid-, and coarse scale MSE decreased in anterior, central, and posterior electrodes, while coarse scale MSE increased in more lateral electrodes. In the third session, only decreases were highlighted in mid- and coarse scales across the scalp.

As brain signal complexity is a property that allows for the formation of and transition between different network configurations [45,46], decreased complexity or MSE may represent a declining need for network transitions during processing, and thus a more simplified neural response. The widespread MSE decreases found here suggest that processing became more efficient with prolonged LDT performance. These MSE decreases could be explained similarly to what theories of repetition suppression suggest [48,50,51,52,53,54], in that the neural response became simplified, thereby optimizing processing and making visual word recognition processes more automatic. Linking these MSE findings with our ERP results and work by Chein and Schneider [6], decreased MSE may in part represent the reduced reliance on control-related processing with increased task efficiency, which allows for a more streamlined neural response during LDT performance.

Interestingly, MSE decreases occurred alongside specific, localized increases. In previous research, higher brain signal complexity has been associated with greater task accuracy [18,19,77,78], more stable RTs [18,77], and with increases in the amount of information available for responding to a stimulus [47,48,77]. With increased information available about a stimulus, there may be engagement of a broader network of regions (i.e., greater repertoire of available responses) that presents as increased signal complexity [45,46], and in turn, this increased repertoire can be thought of as an enrichment of task-related processing [47]. Therefore, our results may indicate that, in addition to the widespread streamlining of processing, an enrichment of processing through greater processing flexibility occurred to a small extent. Further research could examine if these complexity changes might support transfer of processing gains to different, but related, contexts. With decreased complexity, we expect that transfer would not occur due to the discrete specialization of the altered task network. On the other hand, localized increases could indicate greater flexibility in task-related processing, possibly facilitating similar processing in a slightly different context.

Previous work also suggests that complexity aids in the transition from one network configuration to another and enables the exploration of multiple networks to generate optimal responses [79,80]. In this context, it could be expected that MSE changes associated with increased neural efficiency would precede other brain changes (i.e., ERP changes in our study), and may be most prominent in electrodes that later show task-related changes. However, we found that MSE and ERP changes occurred concurrently, and that MSE effects were widespread and only partially overlapped with ERP effects. This fits with a key finding from fMRI work, showing that the spatial distribution of mean-based effects cannot be used to predict the location of variability effects [81,82]. Although this lack of overlap underscores the complementarity of complexity and ERP metrics, it also highlights that MSE effects cannot be interpreted based on the same literature and according to the same logic as ERP effects. In addition, it is possible that a finer temporal resolution is needed to detect a potential spatio-temporal relationship between MSE and ERP effects. In either case, more testing is required to understand the link between signal complexity and ERP changes associated with increased efficiency.

There are several limitations to the current study. In our experiment, our sample size was smaller than that used in many behavioural studies, such as the BLP [2]. Notably, we were able to replicate the behavioural results of the BLP with our sample, and our sample size is consistent with other ERP studies. We also included a large number of trials in our analyses (approximately 1000 trials per participant, per EEG session). Given the within-subjects design of our experiment, this helps to ensure adequate statistical power of our design [83] and adds to the robustness of the effects identified. Further, we did not collect response times during the LDT blocks completed out of the lab as we judged that differences in internet connection speeds would have affected the accuracy of response time measurements. Though the response time measurements taken during the in-lab EEG sessions provided evidence of an increase in efficiency (i.e., decreased LDT response time observed across the sessions), more could be learned from examining the response time changes in all blocks of the experiment.

In addition, we have drawn conclusions regarding neural adaptation and increased efficiency following prolonged performance of the LDT. The LDT is a specific word recognition task, but it also requires several other cognitive processes, such as attention and stimulus evaluation, and thus can act as a model for investigating changes in neural efficiency. Certainly, in order to evaluate the generalizability of our findings, future research should examine changes in complexity after prolonged performance on tasks involving different cognitive domains. Finally, in the current study participants completed approximately 16 h of LDT over a period of one week, and given that the strongest behavioural and neural changes occurred within the first session (approximately 45 min of task performance), future research could make use of a shorter task period to make participant requirements more feasible.

## 5. Conclusions

Despite evidence that visual word recognition is an efficient process for skilled readers [1], the findings of the current study suggest experience-driven performance alterations can still occur. The present findings go beyond those of previous studies [2,3] to show that neural efficiency increases with prolonged performance of the LDT. Our findings suggest increased efficiency occurs through a shift from a reliance on control-related processing, which was prominent early in task performance, to a reliance on domain-specific processing that better supports task-related processing [6]. In addition, the changes in signal complexity suggest that with prolonged performance, processing became more automatic and optimized. Interestingly, the reduction in RT that was observed behaviourally was supported primarily by neural changes that increased processing efficiency, but also by specific changes that represent an enrichment of processing, which should be a focus for future research. Overall, the current findings add to our understanding of how task-related processing becomes more efficient after prolonged performance, in terms of specific processing changes and alterations in brain dynamics. Our findings will also be useful for visual word recognition researchers, as they work towards developing models that are more neurally plausible.

## Figures and Tables

**Figure 1 entropy-23-00304-f001:**
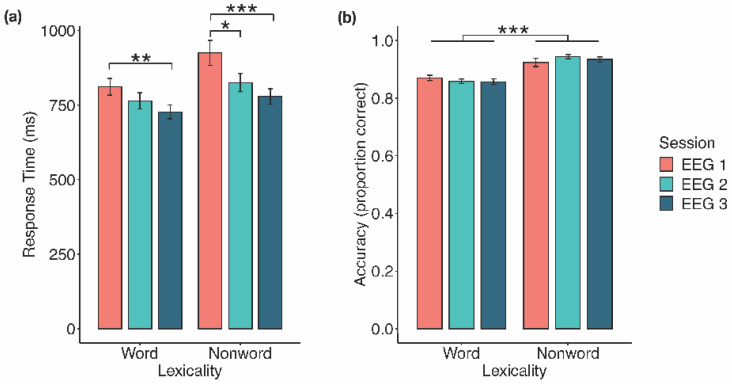
Behavioural effects of prolonged LDT performance. (**a**) Response time decreased significantly for both words and nonwords from Session 1 to Session 3, and from Session 1 to Session 2 for nonwords. (**b**) Responses to nonwords were significantly more accurate than to words across EEG sessions. Error bars represent 1 standard error. * *p* < 0.05, ** *p* < 0.01, *** *p* < 0.001.

**Figure 2 entropy-23-00304-f002:**
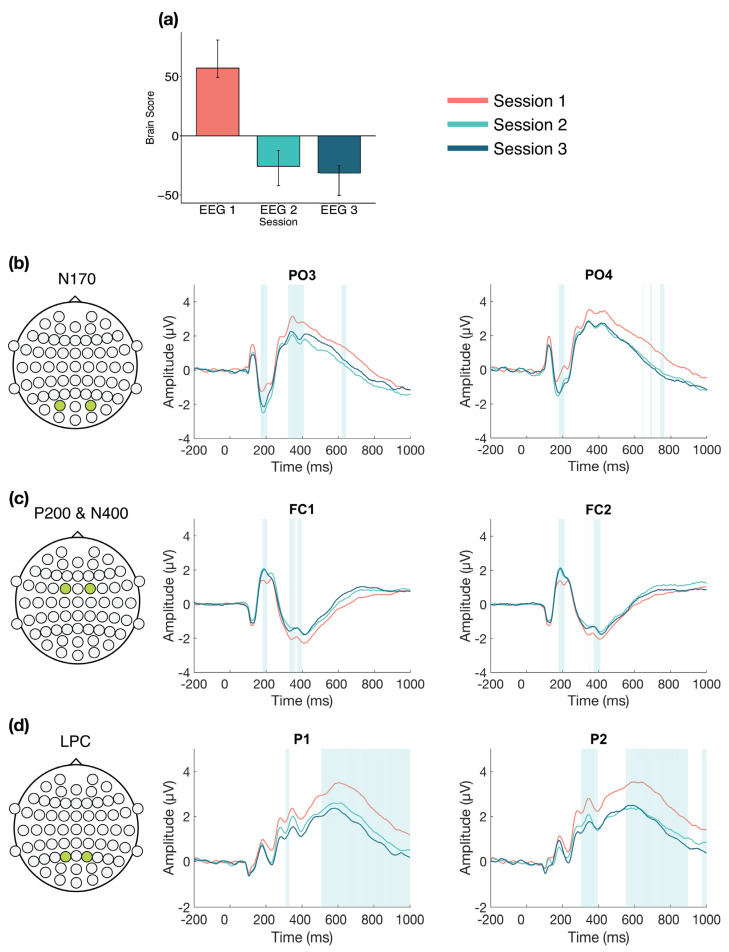
Across session ERP results. Electrodes shown are representative of each effect. Highlighted areas in the ERP waveforms indicate the time points when the difference was reliable, as determined by bootstrap resampling. (**a**) Task PLS analysis differentiated between Session 1 versus Sessions 2 and 3. Error bars represent 95% confidence intervals. (**b**) N170 showing more negative amplitude in later sessions. (**c**) P200 (first highlighted area) and N400 (second highlighted area) showing more positive amplitude in later sessions. (**d**) LPC showing less positive amplitude in later sessions. Head plots in (**b**)–(**d**) indicate locations of electrodes shown.

**Figure 3 entropy-23-00304-f003:**
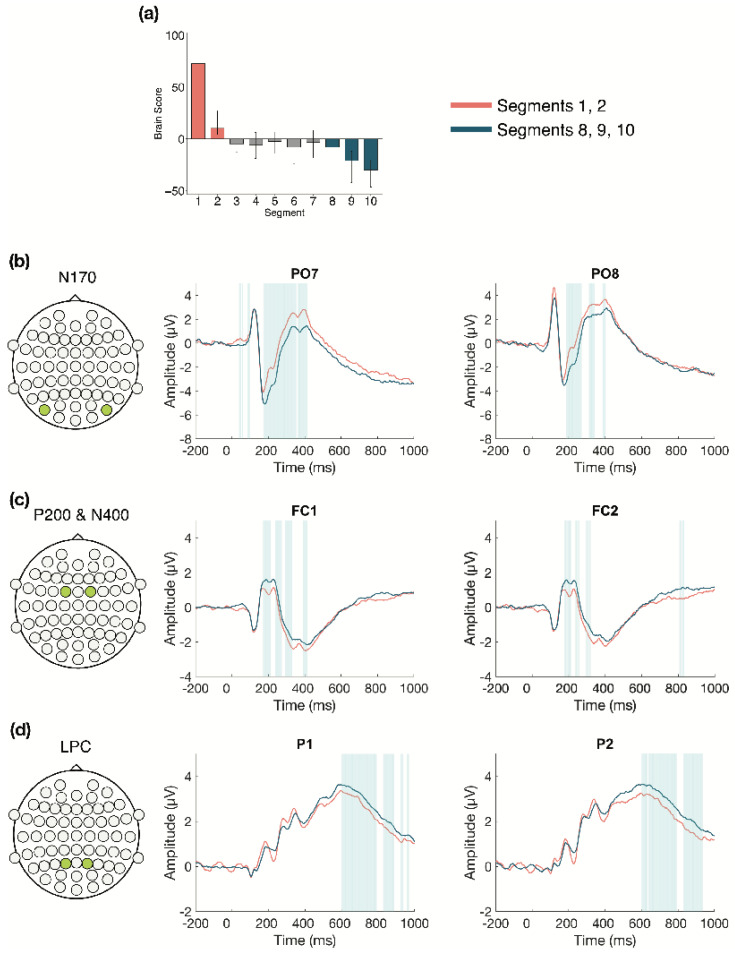
Within session ERP results for Session 1 (LV 1). Electrodes shown are representative of each effect. Highlighted areas in the ERP waveforms indicate the time points when the difference was reliable. (**a**) Task PLS analysis differentiated between the beginning and end of Session 1. Error bars represent 95% confidence intervals. (**b**) N170 showing more negative amplitude later in the session. (**c**) P200 shown in left and right panels, with more positive amplitude later in the session; N400 shown in left panel only, which had left-lateralized effects, with more positive amplitude later in the session. (**d**) LPC showing more positive amplitude later in the session. Head plots in (**b**)–(**d**) indicate locations of electrodes shown.

**Figure 4 entropy-23-00304-f004:**
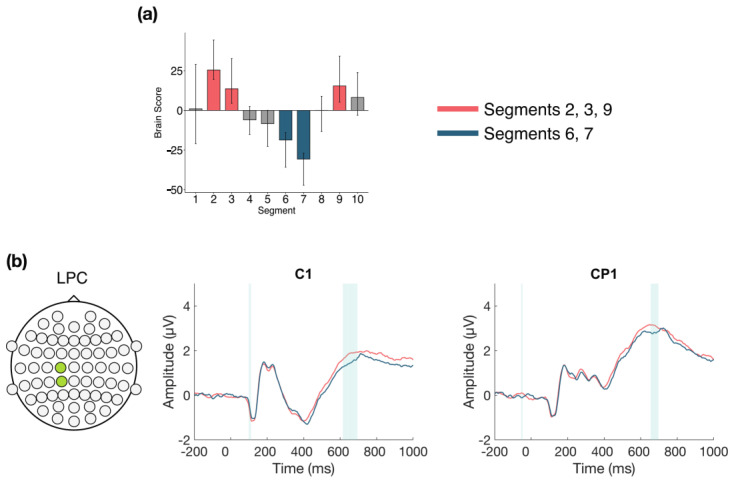
Within session ERP results for Session 1 (LV 2). Highlighted areas in the ERP waveforms indicate the time points when the difference was reliable. (**a**) Task PLS analysis differentiated between the beginning and end compared to the middle of Session 1. Error bars represent 95% confidence intervals. (**b**) LPC showing more negative amplitude in the middle in the session. Electrodes shown are representative of the effect, which was left-lateralized. Head plot indicates locations of electrodes shown.

**Figure 5 entropy-23-00304-f005:**
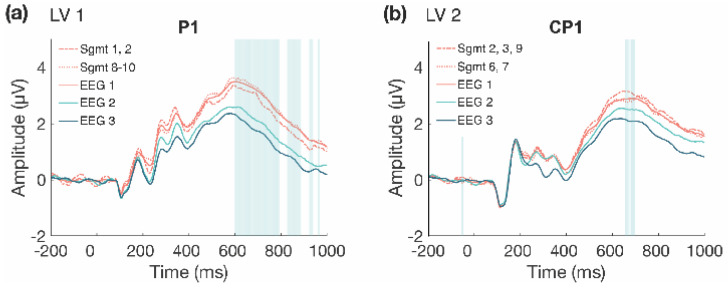
Comparison of LPC waveforms within Session 1. LPC waveforms for LV 1 (**a**) and LV 2 (**b**) within Session 1, with across session waveforms included to depict amplitude change within Session 1 compared to the average amplitudes across the three EEG sessions. (**a**) LPC amplitude is lower at the beginning of the session (dashed line), but still increased compared to Sessions 2 and 3, and then is increased at the end of Session 1 (dotted line). (**b**) Initially, LPC amplitude is increased (dashed line) and then decreases in the middle of the session (dotted line), although the amplitude remains greater than the average amplitude in Sessions 2 and 3, and then increases again at the end of Session 1 (dashed line).

**Figure 6 entropy-23-00304-f006:**
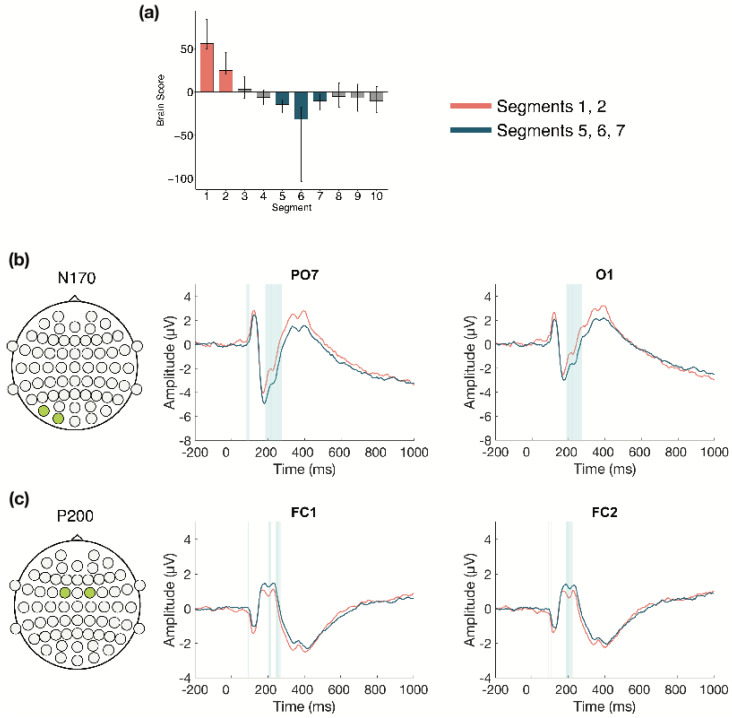
Within session ERP results for Session 2 (LV1). Electrodes shown are representative of each effect. Highlighted areas in the ERP waveforms indicate the time points when the difference was reliable, as determined by bootstrap resampling. (**a**) Task PLS analysis differentiated between the beginning and middle of Session 2. Error bars represent 95% confidence intervals. (**b**) N170 showing more negative amplitude later in the session. (**c**) P200 showing more positive amplitude later in the session. Head plots in (**b**,**c**) indicate locations of electrodes shown.

**Figure 7 entropy-23-00304-f007:**
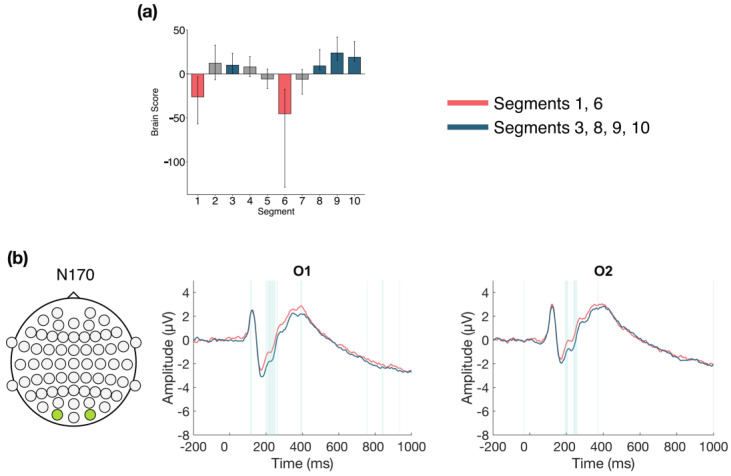
Within session ERP results for Session 2 (LV 2)**.** Electrodes shown are representative of the effect. Highlighted areas in the ERP waveforms indicate the time points when the difference was reliable. (**a**) Task PLS analysis differentiated between the beginning (Segment 1) and middle (Segment 6) of Session 2 compared to the early middle (Segment 3) and end (Segments 8, 9, and 10) of Session 2. Error bars represent 95% confidence intervals. (**b**) N170 showing more negative amplitude during Segment 3 and later in the session, compared to the beginning and middle of the session. Head plot in (**b**) indicates locations of electrodes shown.

**Figure 8 entropy-23-00304-f008:**
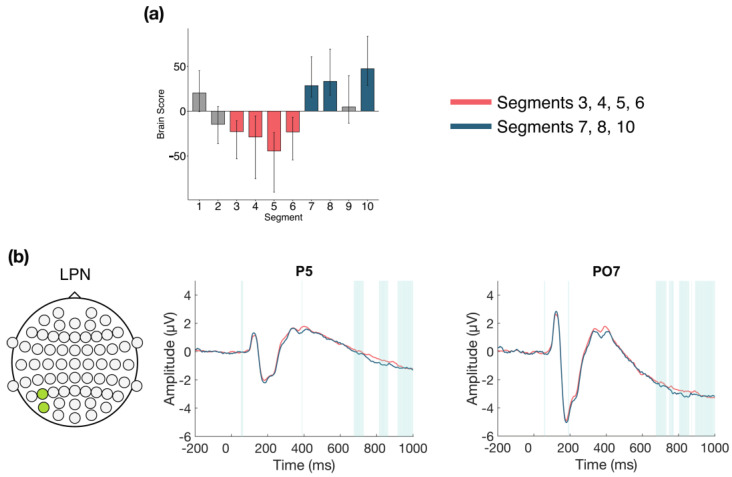
Within session ERP results for Session 3 (LV 1). Electrodes shown are representative of the effect. Highlighted areas in the ERP waveforms indicate the time points when the difference was reliable. (**a**) Task PLS analysis differentiated between the middle and end of Session 2. Error bars represent 95% confidence intervals. (**b**) Negative slow wave, similar to the late posterior negativity (LPN), showing more negative amplitude later in the session. Head plot indicates locations of electrodes shown.

**Figure 9 entropy-23-00304-f009:**
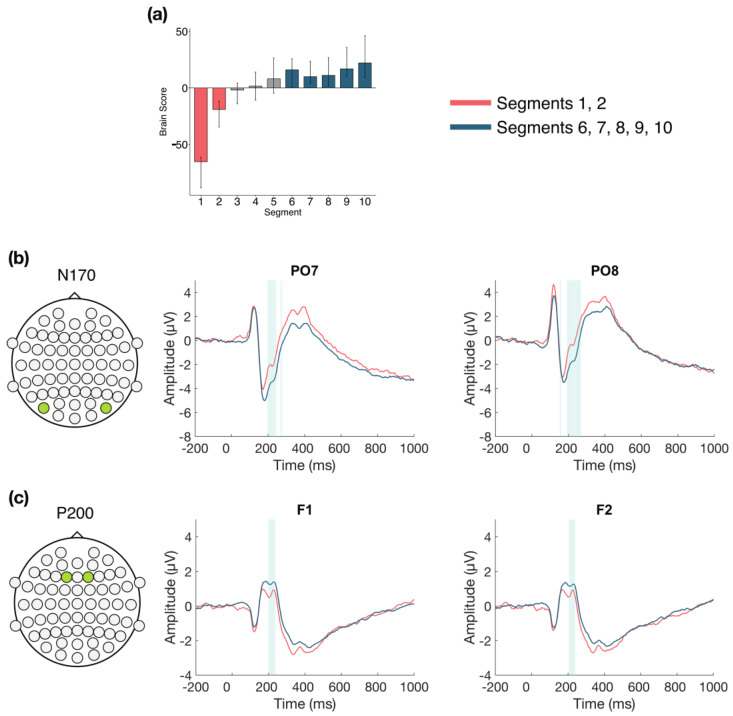
Within session ERP results for Session 3 (LV 2). Electrodes shown are representative of each effect. Highlighted areas in the ERP waveforms indicate the time points when the difference was reliable. (**a**) Task PLS analysis differentiated between the beginning and end of Session 3. Error bars represent 95% confidence intervals. (**b**) N170 showing more negative amplitude later in the session. (**c**) P200 showing more positive amplitude later in the session. Head plots in (**b**,**c**) indicate locations of electrodes shown.

**Figure 10 entropy-23-00304-f010:**
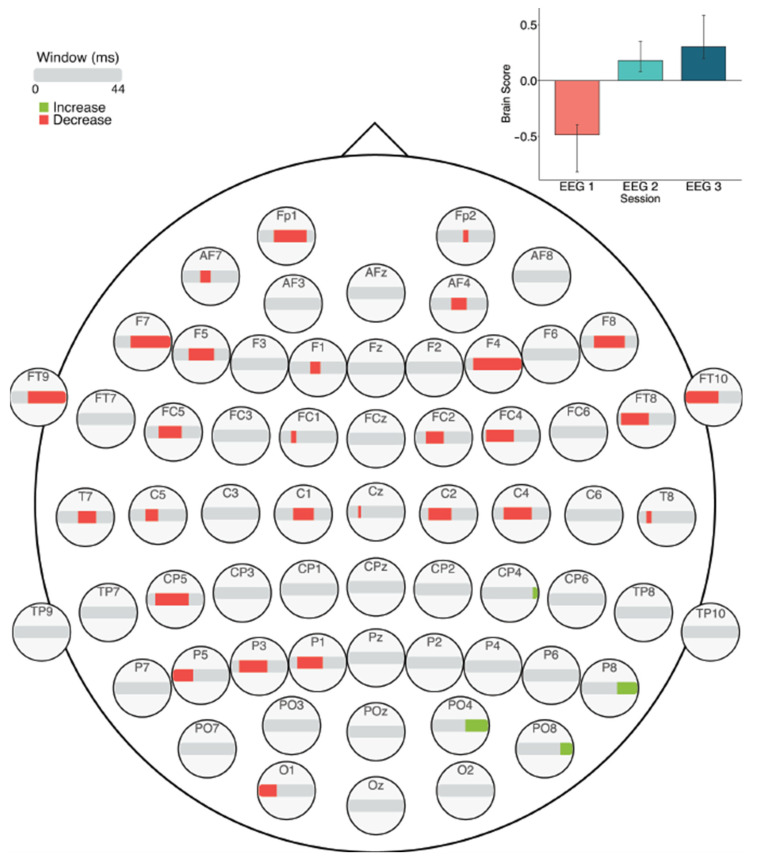
MSE results across sessions. Bar colour indicates the direction of the effect, with red indicating decreases, green indicating increases, and grey indicating no reliable effect. The plot in the top right indicates the differences in MSE that were found between Session 1 versus Sessions 2 and 3. Error bars represent 95% confidence intervals.

**Figure 11 entropy-23-00304-f011:**
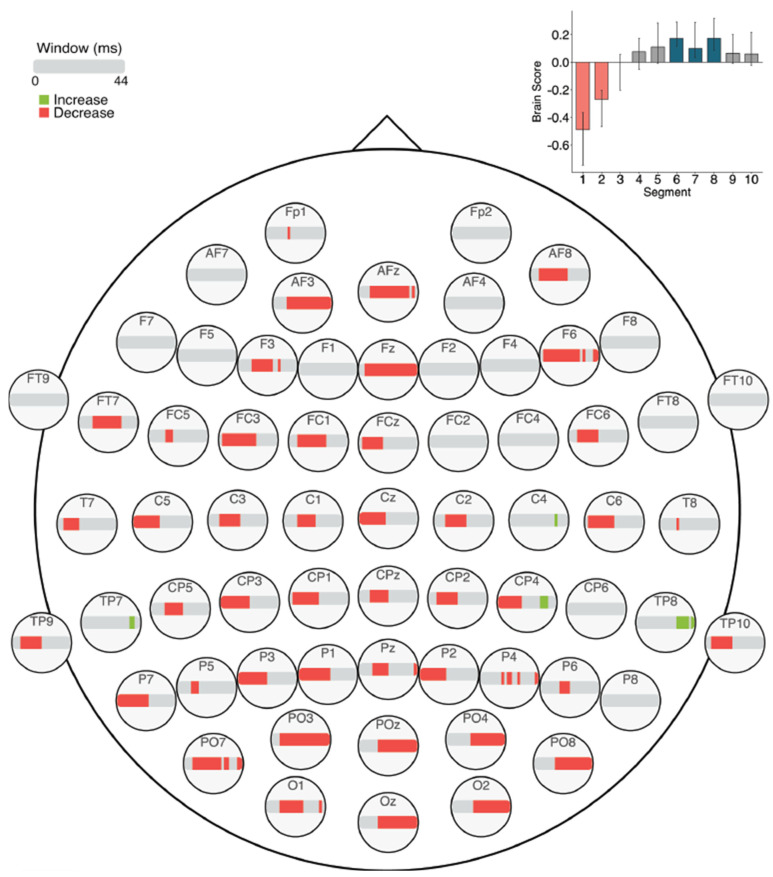
MSE results within Session 1. Bar colour indicates the direction of the effect, with red indicating decreases, and grey indicating no reliable effect. The plot in the top right indicates the differences in MSE that were found between the beginning versus middle/end of Session 1. Error bars represent 95% confidence intervals.

**Figure 12 entropy-23-00304-f012:**
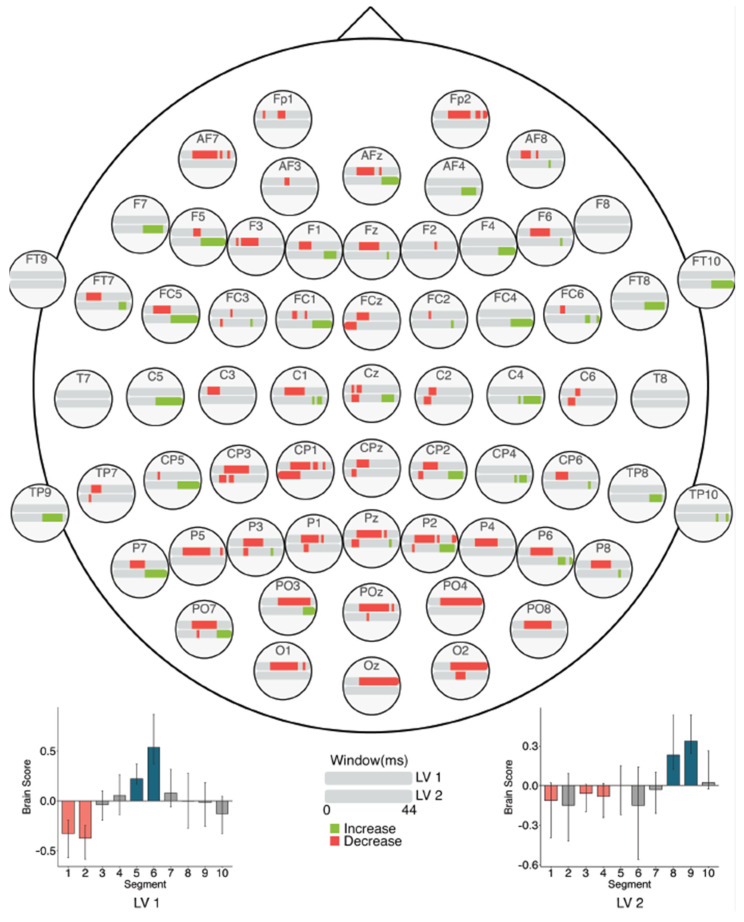
Within session MSE results for Session 2 (LVs 1 and 2). Bar colour indicates the direction of the effect, with red indicating decreases, green indicating increases, and grey indicating no reliable effect. The top bar at each site represents LV 1. The plot at the bottom left indicates the differences in MSE in LV 1 that were found between the beginning and end of Session 2. The bottom bar at each site represents LV 2. The plot at the bottom right indicates the differences in MSE in LV 2 that were found between the beginning and end of Session 2. Error bars represent 95% confidence intervals.

**Figure 13 entropy-23-00304-f013:**
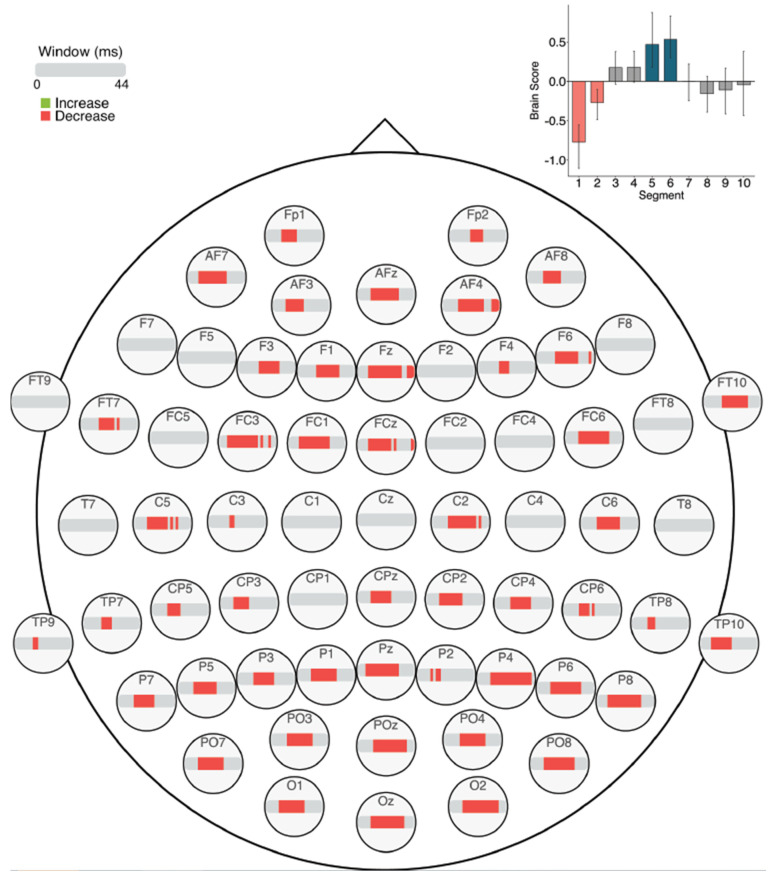
Within session MSE results for Session 3 (LV 1). Bar colour indicates the direction of the effect, with red indicating decreases, and grey indicating no reliable effect. The plot in the top right indicates the differences in MSE that were found between the beginning and middle of Session 3. Error bars represent 95% confidence intervals.

**Table 1 entropy-23-00304-t001:** LDT response time and accuracy for word and nonword trials.

	Response Time (ms)		Accuracy (Proportion Correct)	
	*M*	*SD*	*M*	*SD*
Words				
Session 1	811.16	123.41	0.87	0.04
Session 2	763.96	120.24	0.86	0.04
Session 3	727.24	103.49	0.86	0.04
Nonwords				
Session 1	924.97	185.92	0.92	0.07
Session 2	824.87	136.99	0.94	0.03
Session 3	778.75	115.08	0.93	0.04

Note: *N* = 20. LDT is lexical decision task.

## Data Availability

The behavioural and EEG datasets generated and analyzed during this study, the scripts used to analyze the behavioural and EEG data, and the stimuli used during the study are available on the Open Science Framework website: https://osf.io/da478/, accessed on 28 December 2020.

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
