# Peer review of "Increased Neural Efficiency in Visual Word Recognition: Evidence from Alterations in Event-Related Potentials and Multiscale Entropy"

_entropy, 2021, doi:10.3390/e23030304_

Round 1

Reviewer 1 Report

Paper deals with important task. Authors investigated neural changes during the visual word recognition.

Paper has great practical value, but authors should clearly show the scientific novelty of this paper.

Paper has a logical structure, all necessary sections. 

Suggestions:

  1. It would be good to add point-by-point the main contributions in the end of the Introduction section
  2. It would be good to add the reminder of this paper
  3. Conclusion section should be extended using: 1) limitations of the proposed research; 2) prospects for the future research.
  4. A lot of references are outdated. Please add 3-5 years old papers from high-impact journals.

Reviewer 2 Report

This article explores the neural changes that take place as participants become more efficient at completing lexical decisions.   While the article is generally well written and competently executed, I have some concerns about the exposition of the topic. which feels a bit  "inside-baseball".

  1.  My sense is that *Entropy* is not a journal where word recognition researcher usually publish papers, and the readership might not be totally familiar with why should we even care about the LDT, which seems like such contrived task. Hence, some explanation is needed.  (see, for example, Sophie Leroy's paper, "why is my work so hard?")
  2. The characterization of the LDT requiring "thought and attention" early on seems a little loose.   What exactly do the authors mean?  It is not a deliberative decision like "what car should I buy?" so what is the meaning of "thought"?  Along the same lines, some attention needs to be deployed throughout the duration of the experiment.    The fact that frequency effects are present, for example, demonstrate that word-ness does not simply pop-out.
  3. Of course, this would be an editorial decision and I do not want to overstep my bounds as a reviewer, but it seems to me that some of the behavioral data, and even the component based analyses can be put in an appendix, and they authors should focus on the "why do we care", and the multi scale entropy questions.  Just to give an example, the difference in the RT for words v nonword has really no theoretical interest (it depends on the specific words and nonword); I realize that the analysis is included for completeness, but it does not add much to the main innovation of the paper.
  4. The corollary of the previous point, is that the implications of the MSE analyses seem a bit light.  What are the consequences for word recognition models? for lexical decision models?  for reading research?

Reviewer 3 Report

The paper is well written and well structured. The results are clearly presented and are considered suitable for publication.
